# Defining the Connotations of Oral Health Literacy Using the Conceptual Composition Method

**DOI:** 10.3390/ijerph20043518

**Published:** 2023-02-16

**Authors:** Zhiqiang Tian, Yanjun Wang, Yang Li, Jiao Lu, Li Song, Ling Ding, Xinyu Guo, Jianzhong Zheng

**Affiliations:** 1Shanxi Bethune Hospital, Shanxi Academy of Medical Sciences, Tongji Shanxi Hospital, Third Hospital of Shanxi Medical University, Taiyuan 030032, China; 2School of Health Management, Shanxi Technology and Business College, Taiyuan 030036, China; 3Comprehensive Service Center of Shanxi Medical and Health Institutions (Shanxi Province Blood Center), Changfeng Street, Taiyuan 030006, China; 4School of Public Policy and Administration, Xi’an Jiaotong University, Xi’an 710049, China; 5School of Public Health, Shanxi Medical University, South Xinjian Road, Taiyuan 030001, China

**Keywords:** oral health literacy, conceptual model, conceptual composition

## Abstract

Due to advancements in research, the concept of oral health literacy (OHL) has become rich in connotations, with over 250 definitions present in the literature and government and organizational reports. The diversity of OHL definitions and connotations not only produces conflicting results but also limits the production of accurate OHL measurement and assessment tools while simultaneously hindering the construction of health literacy intervention policies. To clarify the connotations of OHL and establish a scientific basis for evaluation, we conducted a systematic review, searching and analyzing the literature related to the conceptual connotations of OHL. Additionally, we extracted basic, methodological, and OHL conceptual connotation information from the literature. With reference to the review framework, we classified the conceptual connotations of OHL into antecedents of OHL, the core of OHL, mediators, and outcomes of OHL. The comprehensive conceptual connotations of OHL were obtained through a systematic review and concept mapping based on the related literature. Our analysis revealed that the antecedents of OHL can be classified in two categories: personal factors and external factors. The core conceptual connotations of OHL include three core dimensions (with 16 subdimensions): (1) basic skills—literacy, reading comprehension, numeracy, hearing, oral expression, communication, and knowledge; (2) information-related abilities—information acquisition, information understanding, information communication, information evaluation, information utilization, and information decision-making; and (3) oral health maintenance abilities—interpersonal skills, self-regulation, and goal achievement. The mediator of these connotations is oral health behaviors, with oral health being the result of OHL. This study further clarifies the conceptual connotations of OHL, serving as a reference for future OHL-related studies.

## 1. Introduction

Health literacy refers to the ability of individuals to access and understand basic health information and services and to use them to make sound decisions to maintain and promote their health [1]. Oral health literacy (OHL) is a type of health literacy first introduced by Healthy People 2010 in 2000 [2]; it is one of the main determinants of oral health and refers to the extent to which individuals have the ability to access, process, and understand basic oral health information; access and understand basic oral health services; and make appropriate health decisions regarding basic oral health information and services [3]. This concept is widely used, with many researchers having drawn on the abovementioned definition to conduct studies; for example, the American Dental Association (ADA) demonstrated that limited OHL is an obstacle to the prevention, diagnosis, and treatment of oral diseases [4]. Oral diseases not only affect the function of the oral organs but also often affect the health of the whole body. For example, children with more decayed teeth are often thin, and serious decayed teeth affect the growth of children. Paradentitis can cause arthritis, endocarditis, nephritis, and other diseases. Research has shown that individuals who have fewer sources of oral health information—for example, due to personal education level, Internet access, and so on, a subset of health literacy skills—are more likely to fail to show up for dental appointments [4]. Moreover, the National Institute of Dental and Craniofacial Research (NIDCR) recognizes that the generally low level of OHL in the population is a major factor contributing to disparities in oral health among populations [5]. Therefore, OHL is considered crucial for promoting oral health and preventing oral diseases [6]. This is due to the interplay between culture, society, health systems, education systems, language, and oral health. In fact, OHL is becoming a new determinant of oral health, and the richness and complexity of the concept have been augmented through ongoing research, resulting in over 250 definitions in the literature and government and organizational reports [3]. Past researchers have concluded that OHL should be regarded as a multidimensional concept with dynamic characteristics, and its connotations include a variety of abilities and skills related to oral health, such as cognition, reading, comprehension, calculation, communication, listening, decision making, and knowledge [3]. Diverse OHL concepts not only tend to lead to contradictory results [7,8] but also limit the availability of accurate OHL measurement and assessment tools while simultaneously hindering the development of health literacy intervention policies. To the best of our knowledge, there are very few studies that have used scientific methods to define OHL. Therefore, the aim of this study was to comprehensively review the conceptual content of OHL using the previous literature, extract and organize its conceptual connotations, and provide a foundation for future OHL measurement tools and research.

## 2. Research Methodology

The systematic review method was used to retrieve and screen studies related to the connotations of OHL, while the Checklist of Critical Appraisal Skills Programme (CASP) was employed to evaluate the quality of the studies. Relevant information was extracted from prior studies and article-based methodological reviews. Referring to the methods in the literature [9], the conceptual connotations of OHL were divided into antecedents, a core, mediators, and outcomes. Content in the included literature related to the OHL core was extracted, and the concept mapping method was employed to synthesize the OHL cores defined in different studies and clarify the key concepts. The following corresponding steps were employed: (1) identifying key concepts, (2) identifying the order in which concepts were arranged, (3) specifying links to key concepts, and (4) ensuring that the overall order clarified the connotations of OHL.

### 2.1. Search Strategies and Screening Criteria

The databases searched in this study included PubMed, Web of Science, Scopus, Cochrane Library, ScienceDirect, and Springer. A combination of OHL and its conceptualization was adopted as follows: “oral health literacy”, “dental health literacy”, “OHL”, “literacy in dentistry”, “oral”, “dental”, “health”, “literacy”, “definition”, “concept”, “defining”, “dimension”, “dimensionality”, ”frame”, “framework”, “conceptual framework”, “analysis”, “theory”, “qualitative”, and “model”. For these databases, the search period spanned 1 January 1990 to 1 February 2020. Full-text screening of the retrieved studies was used for the inclusion criteria. The objective of the study was to conceptualize OHL and interpret and extend its concept. Exclusion criteria for the studies were as follows: (1) the study was not related to the OHL concept; (2) the OHL concept was not explained; (3) the study used the existing OHL concept without further extension; and (4) the study was in a language other than Chinese or English. Where there were disagreements about the studies to be included, two researchers deliberated, and if agreement could not be reached, then the final decision was made by a third investigator. The search process is shown in Figure 1.

### 2.2. Conceptual Composition

Conceptual composition refers to a concept map that organizes and represents the tools of knowledge; such knowledge usually pertains to a topic with different levels of concepts or propositions in a box or circle. A variety of connections link the relevant concepts and propositions to form a concept or proposition network on the topic. Thus, the learner’s knowledge structure and understanding in relation to a topic can be characterized visually. A conceptual diagram of a research object includes four basic elements: nodes, connections, hierarchies, and propositions. The conceptual composition method was used in this study to summarize and analyze the different research results and scientifically expound the conceptual connotations of OHL. Mapping of the conceptual composition was performed in the following four steps.

#### 2.2.1. Clarification of Key Concepts

This step involved reading the relevant literature, understanding the depth and breadth of the research problems, selecting key concepts and related concepts within a specific knowledge range, and listing those concepts. As per the goal of the present study, 20 experts in the fields of oral medicine education and oral medical practice with intermediate and senior professional titles (work experience of more than eight years) were invited to participate in this study. Following the implementation of the Delphi method, two rounds of expert consultations were conducted via e-mail. The consultation letter and relevant background information were provided to the experts at the time of the consultation, along with a checklist requiring the experts to determine whether the key concepts for the topic were included. The list also asked them to make comments and suggestions for modifications and explain the basis of their judgments. The key concepts for the topics were obtained by the research group after the literature review and expert discussions. On this basis, the results of the first round of consultations were sent back to the experts and the second round of Delphi expert consultations were conducted to solicit their opinions. In both rounds of consultations, the experts were asked to score each key concept according to its importance on a scale of one to seven points to average the scores for each key concept (the higher the score, the greater the importance). The scores were divided by the total score to obtain weight coefficients.

#### 2.2.2. Order of the Concepts

The weight coefficients of the key concepts obtained after two rounds of expert consultations were sorted. Concepts with large weight coefficients were placed in the center of the map and concepts with small weight coefficients or materializations were placed around them, thus forming a hierarchy of concepts.

#### 2.2.3. Links to Key Concepts

Interrelated concepts were connected with lines, and notes regarding the logical relationships between them were included. Different knowledge nodes were organically linked through key concepts to form the conceptual composition.

#### 2.2.4. Overall Improvement

The third round of expert consultations will be carried out using the initially constructed concept map, and the opinions and suggestions of experts will be obtained. Accordingly, the concept map will be further enhanced and modified to achieve smooth concept relationships and a perfect composition.

The flowchart for the analysis of the conceptual connotations of OHL is shown in Figure 2.

## 3. Results

### 3.1. Basic Information Regarding Inclusion

Table 1 summarizes the basic information for the 17 included articles [2,3,8,10,11,12,13,14,15,16,17,18,19,20,21,22,23]. Among them, 3 papers explained the definition or concept of OHL, and 14 focused on its definition while also exploring its moderators and outcomes. Ten studies targeted the general population [2,3,10,14,15,16,17,19,21,23], and seven focused on special populations [8,11,12,13,18,20,22], including low-income adults [8]; low-income women, infants, and children [11]; low-income patients undergoing their first pregnancy [12]; older adults (65 and older) [13,22]; rural adults [18]; and stomatologists [20]. Among all the OHL definitions, the one most commonly used was that proposed as part of the 2000 Healthy People 2010 agenda. In 2005, the NIDCR and Institute of Medicine (IOM) provided a definition of health literacy for use in the field of dentistry, expanding the definition of OHL and emphasizing functional OHL and the ability to apply the corresponding knowledge to make decisions related to oral health. The definition also suggested that oral health information can be communicated in a variety of ways [10]. Van Wormer et al. [18] adopted the comprehensive conceptual model of health literacy proposed by Sorensen et al. in the context of OHL, which synthesizes the main dimensions of health literacy developed in the previous model (e.g., access to and understanding and application of health information). The moderators and outcomes of health literacy were included. Furthermore, Spivakovsky et al. [23] argued that OHL-related knowledge is influenced by socioeconomic characteristics, beliefs, self-efficacy, and previous experiences and that it ultimately affects oral health outcomes. Almost all OHL definitions identified the core as a skill or competency, and some studies expanded OHL-related knowledge [8].

### 3.2. Methodological Information Regarding Inclusion

The methods found in the literature review included (1) cross-sectional methods, face-to-face interviews, and randomized controlled trials and (2) research methods based on pure theory [2,23] and a panel discussion regarding the dimensions of OHL [8]. These two types of studies explored the definition of OHL based on the authors’ experience or existing OHL studies (see Table 2 for details).

### 3.3. Antecedents of OHL

Among the 17 included studies, 15 explored the causes influencing the generation of OHL [23]. According to the aforementioned review framework, researchers believe that the two factors affecting the generation of OHL can be categorized as antecedents of OHL, and these two factors are summarized as follows.

#### 3.3.1. Personal Factors

General characteristics: age, race, gender, education level, income, cultural background, language differences, socioeconomic status, and so on;Personality characteristics: attitude, beliefs, psychological characteristics, self-efficacy, a sense of access, and satisfaction with oral health services.

#### 3.3.2. External Factors

Oral health service providers: communication skills of medical staff;Oral medical service system: structure of the oral health service system, social support, patient education, insurance reimbursement, and so on;Social factors: community services, social environment, education systems, social activities, government officials, the business sector, public and medical libraries, professional and community groups, and public health actions;Personal external factors: family and friends.

### 3.4. The Core of OHL

In total, 11 articles [2,3,8,10,12,13,14,15,16,17,23] defined OHL as comprising individual abilities or skills, including oral expression, reading, writing, and listening, as well as information acquisition, information processing, information comprehension, and information decision-making, which emphasizes functional OHL, including the ability to use the knowledge acquired to make decisions related to oral health. Oral health information is communicated in a variety of ways. Macek (2010) et al. [8] added the conceptual knowledge of oral health to its core dimensions and expanded it to include basic knowledge on oral health and the prevention and treatment of tooth decay, periodontal disease, and oral cancer. Ju et al. [15] made memory and reasoning important parts of the conceptual OHL model. Stein et al. [17] stated that communication and self-management skills are key components of OHL, with these abilities being closely related to the ability to obtain and utilize oral health information. Spivakovsky et al. [23] added resource utilization capabilities that expand the ability to use information (see Table 3 for details).

### 3.5. OHL Mediation and Outcomes

The effects of OHL were analyzed in 7 articles [3,10,18,20,21,22,23], 5 of which suggested that OHL could influence oral health outcomes by altering oral health behaviors and awareness [3,10,18,20,21], while 12 studies concluded that OHL had a direct impact on oral health outcomes [3,8,10,13,14,15,16,17,18,19,20,21]. At the individual level, OHL refers to an individual’s ability to obtain oral health knowledge; improve their acquisition of information; understand, process, apply, and evaluate information and make decisions with it; engage in better doctor–patient communication; and improve oral health and oral health-related quality of life by changing their beliefs, attitudes, behaviors, and self-efficacy. At the group or societal level, OHL is thought to improve outcome indicators, such as quality of care, health satisfaction, and social equity, and facilitate a healthy and harmonious health care environment.

### 3.6. Conceptual Composition Synthesis

Thanks to the combination of the literature review and the three rounds of expert consultations, seven key concepts were generated: personal factors, external factors, basic skills, information-related abilities, oral health maintenance ability, oral health behavior, and oral health status. The specific weight coefficients are shown in Table 4. According to the coefficients, basic skills, information-related competencies, and oral health maintenance competencies, which had the top three weighting coefficients, were classified as the core of the oral health literacy concept. Personal and external factors were the antecedents, oral health behaviors were mediators, and oral health status was the outcome.

### 3.7. OHL Conceptual Connotations

Based on the results for the conceptual composition, OHL’s core conceptual connotations were judged to include three core dimensions (with 16 subdimensions), among which the information-related capability dimension played a bridging and pivotal role (see Figure 3 for details). The core and the subdimensions of OHL constitute the conceptual connotations of OHL.

## 4. Discussion

The concept of OHL was first introduced by Rural Healthy People 2010 [2], and most of the previous studies used their definition or expanded it [11,12,13,16,17,19,21]. However, some studies and reports still exist that use different OHL definitions. In the current study, the relevant contents of the OHL concepts in the research literature were synthesized and summarized, and seven key concepts were extracted. The connotations of these seven key concepts were ranked according to their weights with the assistance of the Delphi method and, finally, the logical order of the seven key concepts was clarified. The core content of OHL comprised three core dimensions (basic skills, information-related competence, and oral health maintenance competence). Basic skills determine the strength of individuals’ information-related abilities, which are the cornerstone of OHL. Information-related competencies measure the overall level of OHL among individuals in terms of information acquisition, comprehension, communication, evaluation, utilization, and decision-making, respectively. The information-related competence dimension plays a bridging and pivotal role for the core dimensions of the conceptual connotations of OHL, while oral health maintenance competence and information-related competence facilitate each other and help people use oral health information effectively. In addition, this study categorized personal and external factors with small weight coefficients as antecedents that influence the acquisition of OHL, in addition to the categorization of oral health behaviors as mediators and oral health statuses as outcomes; that is, OHL can influence oral health status by changing oral health behaviors.

Most of the literature comprises cross-sectional studies and randomized controlled trials with reliable findings, and the three rounds of consultations using the Delphi method (i.e., organizing and summarizing the conceptual connotations) verified the core position of the main concepts and the comprehensiveness of the range of OHL conceptual connotations. This was consistent with the definition of OHL in most of the previous studies [2,3,8,10,11,12,13,14,15,16,17,18,19,20,21,22,23]. Moreover, the current work sorted out the logical relationships between the conceptual connotations of OHL and presented them in a conceptual schema, which is logically clear, easier to understand, and more convenient for scholars conducting OHL research. In particular, scholars who are interested in developing OHL assessment tools can directly draw on this study’s conceptual constructs, the primary- and secondary-order OHL connotations, and the logical relationships to develop questionnaires and establish assessment tools.

## 5. Conclusions

In this study, using the systematic review method, the conceptual contents related to OHL in 17 studies were extracted and summarized, the Delphi method was applied to rank the weights of the conceptual contents, and, finally, a construct comprising seven key OHL concepts was derived. Among these concepts, the antecedents of OHL included personal factors and external factors. The core of OHL comprised three dimensions (with 16 subdimensions) and was mediated by oral health behaviors, while the outcome was oral health status. This study clearly demonstrated the conceptual connotations of OHL and divided its levels and dimensions, presenting the logical relationships between those levels and dimensions and providing a comprehensive, clear, and intuitive reference for the scientific evaluation of OHL and the establishment of related measurement and assessment tools.

## Figures and Tables

**Figure 1 ijerph-20-03518-f001:**
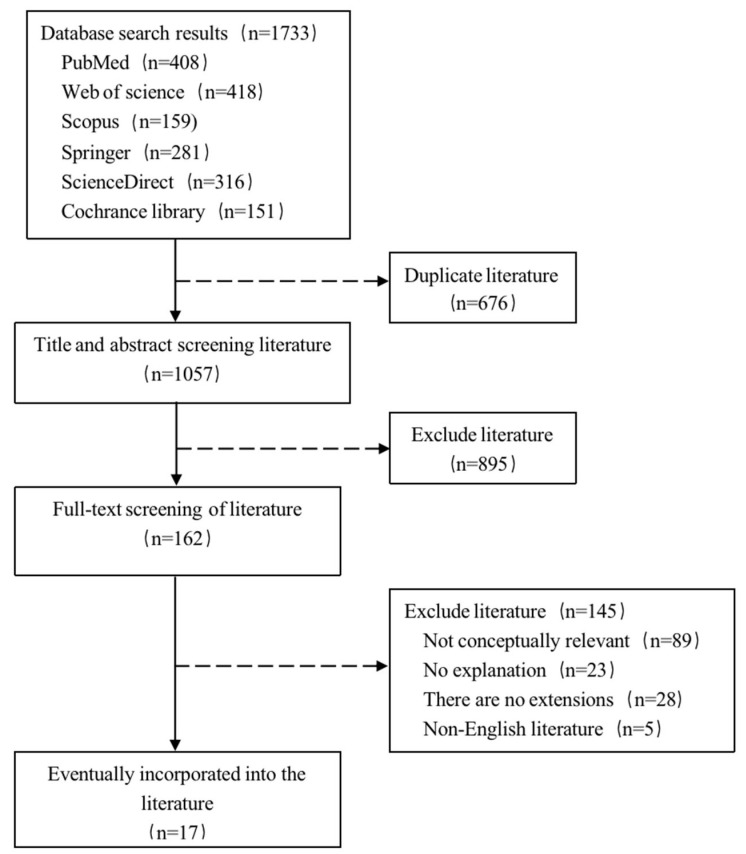
The literature search process.

**Figure 2 ijerph-20-03518-f002:**
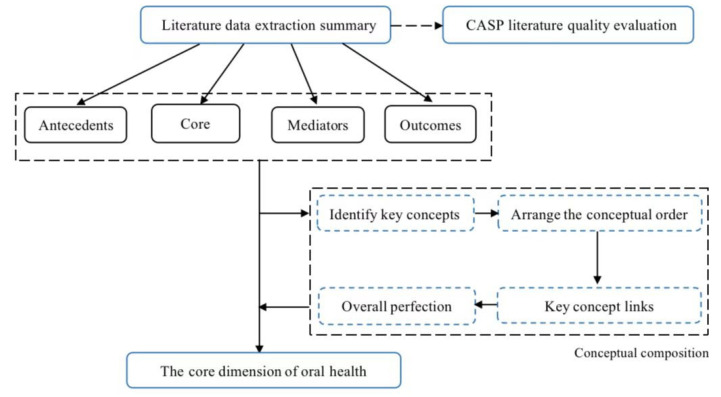
Flowchart for the analysis of the conceptual connotations of OHL.

**Figure 3 ijerph-20-03518-f003:**
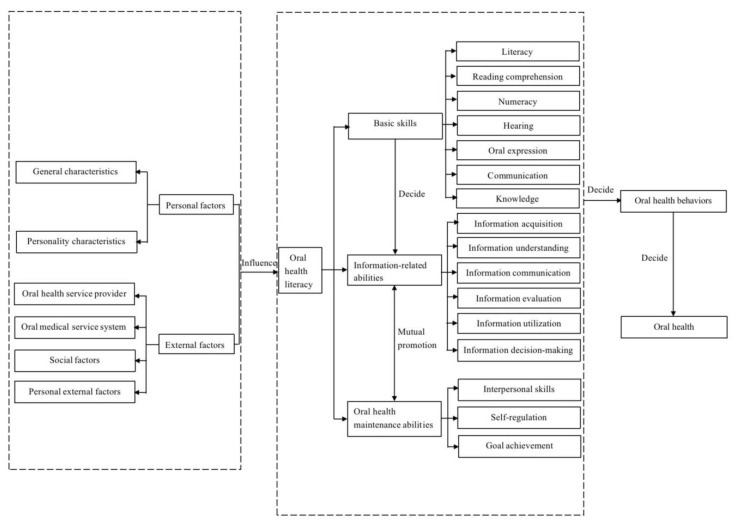
OHL conceptual connotations.

**Table 1 ijerph-20-03518-t001:** Summary of basic information included in the OHL literature.

Serial Number	Author (Year)	Research Objectives	Target Demographics	Oral Health Literacy (OHL) Definition or Interpretation in Relevant Studies
1	U.S. Department of Health and Human Services (2000) [2]	Provide guidance, technical tools, and resources for the development of countries, territories, tribes, and communities to help them implement the 2010 Human Health Plan and other oral health plans	General population	OHL is the degree to which an individual has the ability to access, process, and understand basic oral health information and access and understand basic oral health services needed to make appropriate health decisions
2	Service USPH SH (2005) [10]	Define OHL and provide a framework for studying the relationship between OHL and other intervention points to improve oral health	General population	Individuals can access, process, and understand essential oral health information and access and understand essential oral health services to make appropriate health decisions and take action
3	Macek et al. (2010) [8]	Measure conceptual health knowledge in the context of OHL	Low-income adults	OHL includes four abilities: word recognition, reading comprehension, communication skills, and conceptual knowledge. Oral health conceptual knowledge includes basic knowledge on oral health and the prevention and management of caries, periodontal disease, and oral cancer
4	Lee et al. (2011) [11]	Identify OHL levels for low-income women, infants, and children and explore whether there are racial disparities	Low-income women, infants, and children	OHL is the degree to which an individual has the ability to access, process, and understand basic oral health information and access and understand basic oral health services needed to make appropriate health decisions (the definition proposed by Healthy People 2010 in 2000 was adopted)
5	Horn et al. (2012) [12]	Determine the correlation between OHL levels and oral health knowledge among low-income patients undergoing first-time pregnancies	Low-income patients undergoing first-time pregnancies	Using the definition cited above and proposed by Healthy People 2010 in 2000, it can be concluded that oral health knowledge correlates significantly with OHL levels
6	Kaur et al. (2015) [3]	(1) How do we assess oral health knowledge?(2) What is the relationship between oral health knowledge, oral health outcomes, and access to and satisfaction with dental health services? (3) What interventions are in place for vulnerable populations with low OHL?	General population	Although not mentioned, the moderators and outcomes of the connotations of the concept of OHL were further explained and extended
7	McQuistan et al. (2015) [13]	Understand the level of oral health knowledge in older adults aged 65 and older	Seniors (65 and older)	The definition proposed by Healthy People 2010 in 2000 (cited previously) was adopted
8	Macek et al. (2017) [14]	Extend previous work by using larger study samples and additional outcome variables	Adults	Although not mentioned, the study provides additional support for the effectiveness of the Comprehensive Measure of Oral Health Knowledge (CMOHK). Researchers were encouraged to incorporate knowledge on oral health concepts into their theoretical frameworks, especially regarding beliefs and self-efficacy. Key observational indicators in this study included dental use, self-efficacy, and dental beliefs and attitudes
9	Ju et al. (2017) [15]	Determine the effect of the OHL intervention on OHL-related outcomes among Australian Aboriginal adults living in rural areas	Aboriginal Australian adults living in rural areas	Although not mentioned, the study was partially successful in the functional, targeted improvement of OHL and the OHL-related outcomes of OHL interventions
10	Bridges et al. (2014) [16]	Describe the relationship between caregivers and children’s oral health	Children’s caregivers	As described previously, OHL is the degree to which an individual has the ability to access, process, and understand basic oral health information and access and understand basic oral health services needed to make appropriate health decisions (the definition proposed by Healthy People 2010 in 2000 was adopted)
11	Stein et al. (2018) [17]	Examine the role of the health literacy conceptual model in a clinical stomatology setting	Norwegian-speaking adults	The definition proposed by Healthy People 2010 in 2000 was adopted (cited above)
12	Van Wormer et al. (2019) [18]	Study the relationship between OHL, sociodemographic factors, and several oral health outcomes when rural adults receive integrated medical and dental care services	Rural adults	The comprehensive conceptual model of health literacy proposed by Sorann et al. was adopted and applied to the context of OHL. This conceptual model synthesized the main health literacy dimensions developed in the previous model (e.g., access to and understanding and application of health information, including the antecedents and consequences of health literacy)
13	Mohammadi et al. (2018) [19]	Assess adults’ OHL levels and their associated factors	Adults	The level of an individual’s ability to process and understand basic information about oral health and related services (the definition proposed by Healthy People in 2010 was adopted)
14	Baskaradoss (2018) [20]	Explore the relationship between OHL and oral health	Stomatology patients	Although not mentioned, the study found that subjects with limited levels of OHL had poorer periodontal health. Improving a patient’s OHL level may help improve compliance with physicians’ orders, self-management skills, and overall treatment outcomes
15	Márquez-Arrico et al. (2019) [21]	Analyze the relationship between oral health knowledge and literacy, oral hygiene habits, eating habits, and oral health-related quality of life among adults in Spain	Adults in Spain	The level of an individual’s ability to process and understand basic information regarding oral health and related services (the definition proposed by Healthy People in 2010 was adopted)
16	Tenani et al. (2019) [22]	Explore the effects of OHL and its related factors on oral health dissatisfaction among older adults	Senior citizens	Individuals can access, process, and understand essential oral health information and access and understasnd essential oral health services to make appropriate health decisions and take action (2005 National Institute of Dental and Craniofacial Research (NIDCR) definition of OHL)
17	Spivakovsky et al. (2020) [23]	Introduce the OHL scale for Spanish speakers and understand the factors that affect the oral health of Spanish speakers	Spanish speakers	OHL-related knowledge is influenced by socioeconomic characteristics, beliefs, self-efficacy, previous experiences, etc. and ultimately affects oral health outcomes

**Table 2 ijerph-20-03518-t002:** Summary of methodological information included in the OHL literature.

Serial Number	Author (Year)	Research Methodology
1	U.S. Department of Health and Human Services (2000) [2]	Based on the definition of health literacy and application of that definition to the context of OHL (theoretical construction).
2	Service USPH SH (2005) [10]	Adoption of the definition proposed by Healthy People 2010 in 2000, explaining and extending it (cross-sectional study)
3	Macek et al. (2010) [8]	Panel discussions
4	Lee et al. (2011) [11]	Face-to-face interviews
5	Horn et al. (2012) [12]	Face-to-face interviews
6	Kaur et al. (2015) [3]	Scoping review
7	McQuistan et al. (2015) [13]	Interviews
8	Macek et al. (2017) [14]	Cross-sectional studies
9	Ju et al. (2017) [15]	Randomized controlled trials
10	Bridges et al. (2014) [16]	Face-to-face interviews
11	Stein et al. (2018) [17]	Randomized controlled trials
12	Van Wormer et al. (2019) [18]	Cross-sectional studies
13	Mohammadi et al. (2018) [19]	Cross-sectional studies
14	Baskaradoss (2018) [20]	Cross-sectional studies
15	Márquez-Arrico et al. (2019) [21]	Cross-sectional studies
16	Tenani et al. (2019) [22]	Cross-sectional studies
17	Spivakovsky et al. (2020) [23]	Theoretical construction

**Table 3 ijerph-20-03518-t003:** OHL conceptual model information summary.

Serial Number	Author (Year)	Moderators	Attributes	Mediation	Outcomes
1	U.S. Department of Health and Human Services (2000) [2]		Ability to obtain information, information processing ability, information comprehension skills, information decision-making skills		
2	Service USPH SH (2005) [10]	External factors: health service providers, researchers, educators, policymakers, government officials, the business sector, public and medical libraries, professional and community groups	Reading and writing abilities, listening, oral expression ability, information acquisition ability, information processing ability, information comprehension ability, information decision-making ability	Oral health awareness	
3	Macek et al. (2010) [8]		Text recognition ability, reading comprehension ability, communication skills, decision-making ability Conceptual knowledge: (a) basic knowledge of oral health (b) prevention and management of tooth decay (c) prevention and management of periodontal disease (d) prevention and management of oral cancer		
4	Lee et al. (2011) [11]	Personal factors: race			
5	Horn et al. (2012) [12]	Personal factors: knowledge External factors: cultural and linguistic differences, insurance reimbursement status	Knowledge		
6	Kaur et al. (2015) [3]	Personal factors: oral health knowledge External factors: access to and satisfaction with oral health services	Ability to obtain information, information processing capabilities, information comprehension ability, information decision-making skills	Oral health behaviors	Oral health
7	McQuistan et al. (2015) [13]	Personal factors: income level, age, education, socioeconomic status, oral hygiene practices	Reading ability, writing skills	Oral health awareness	
8	Macek et al. (2017) [14]	Personal factors: beliefs, attitudes	Oral conceptual knowledge, self-efficacy		
9	Ju et al. (2017) [15]	Race, education level, age, occupation, income, social support, culture, language, self-efficacy, beliefs, motivation, media, oral health education	Vision, hearing, language skills, memory, reasoning skills, knowledge, communication skills		
10	Bridges et al. (2014) [16]	External factors: community, public health education, family	Literacy, comprehension ability		
11	Stein et al. (2018) [17]	Changes in personal behavior Public health is universal Public health action	Knowledge, communication skills, self-management ability, motivational ability		Gum condition, oral hygiene
12	Van Wormer et al. (2019) [18]	Personal factors: education level, oral health information sources, age		Oral hygiene practices	Oral hygiene status
13	Mohammadi et al. (2018) [19]	Personal factors: socioeconomic status, ethnicity, annual household income			Quality of life, oral diseases
14	Baskaradoss (2018) [20]	Personal factors: ethnicity, socioeconomic status		Oral health behaviors	Oral health
15	Márquez-Arrico et al. (2019) [21]	Personal factors: education level, age, gender			Oral health-related quality of life
16	Tenani et al. (2019) [22]	Socioeconomic variables			Oral health
17	Spivakovsky et al. (2020) [23]	Socioeconomic characteristics, beliefs, self-efficacy, etc. Previous experience	Oral conceptual knowledge, resource utilization capabilities		

**Table 4 ijerph-20-03518-t004:** OHL key concepts and their weight factors.

Incorporated Concepts	Weight Coefficients
Basic skills	0.309
Information-related capabilities	0.225
Oral health maintenance ability	0.135
Personal factors	0.129
External factors	0.084
Oral health behaviors	0.061
Oral health status	0.057

## Data Availability

Data are contained within the article.

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
