# Peer review of "Defining the Connotations of Oral Health Literacy Using the Conceptual Composition Method"

_ijerph, 2023, doi:10.3390/ijerph20043518_

Round 1

Reviewer 1 Report (Previous Reviewer 1)

Still, please in your introduction, describe more about oral health information that people needs to know. Like, what is affecting oral health? Teeth? Smoking? Anything else? What is a typical oral disease? How is oral health related to physical health? You are giving readers a brief idea with lots of discussion but please explain it more at the beginning with more details. By doing that, the significance of the paper will be emphasized. 

Author Response

  1. Comment: Still, please in your introduction, describe more about oral health information that people needs to know. Like, what is affecting oral health? Teeth? Smoking? Anything else? What is a typical oral disease? How is oral health related to physical health? You are giving readers a brief idea with lots of discussion but please explain it more at the beginning with more details. By doing that, the significance of the paper will be emphasized. 

Response: Thank you very much for your comment and suggestion. We very much appreciate your hard work and consideration. In accordance with your comment, we have added the following content to the introduction ( P2-3): Oral diseases not only affect the function of the oral organs, but also often affect the health of the whole body. For example, children with more decsyed teeth are often thin, and serious decsyed teeth affect the growth of children. Paradentitis can cause arthritis, endocarditis, nephritis and other diseases. Research has shown that individuals who has fewer sources of oral health information, a subset of health literacy skills, are more likely to fail to show for dental appointments, for example, personal education level, Internet access and so on[4]. We hope this modification meets with your approval.

Reviewer 2 Report (Previous Reviewer 2)

The authors provided all corrections required.

Author Response

  1. Comments:The authors provided all corrections required.

Response: Thank you very much for your suggestions. Your care and thoughtfulness are much appreciated.

Reviewer 3 Report (Previous Reviewer 3)

Thanks to the Authors for their corrections. The paper looks much better now.

Author Response

  1. Comments:Thanks to the Authors for their corrections. The paper looks much better now.

Response: Thank you very much for your suggestions. Your care and thoughtfulness are much appreciated.

This manuscript is a resubmission of an earlier submission. The following is a list of the peer review reports and author responses from that submission.

Round 1

Reviewer 1 Report

In this paper, researchers described the development of a conceptual model related to oral health literacy. The paper contains a lot of information with useful results that would benefit the public health as well as other researchers in this field. However, this paper still needs improvements. Please check the following comments:

1.     Please enrich your introduction with more detailed information. For instance, in introduction, line 4, you mentioned that oral health is related to physical health. Please list some examples of how oral health affect physical health. This will emphasize the importance of oral health, and readers will know it is highly related to their daily lives. Another thing you could talk about oral health is what is the major factors affecting oral health among Chinese people? Smoking? Teeth brushing and related stuff? Or something else? Currently, your introduction is providing readers a broad idea. More details are definitely needed.

2.     Please also improve references. Usually, after a solid statement, you provide references. Please go over your paper and add references accordingly. 

3.     I would suggest you make a table and list the factors on page 8. In this way, readers will get the ideas easily and efficiently. 

4.     You have delineated the methods and results in a very detailed way and discussed nicely based on what you found. However, at the beginning of the paper, also include the title of the paper, it seems like you are focusing on Chinese people. However, you don’t really talk about Chinese people and their oral health specifically overall in your paper. The suggestion is to build a link between your current stuff and Chinese people, especially in results and discussion part. 

Reviewer 2 Report

Thank you for the submission. The title is acceptable and relevant to the context. It is undeniable the importance of oral health literacy and its impact on health promotion.

-      This section below is redundant as it features 3 OHL definitions which are interchangeable. I suggest the adoption of only one in order to make the introduction more concise and objective:

      OHL is a main determinant of oral health and refers to the degree to which an individual has the ability to access, process, and understand basic oral health information and services needed for making appropriate health decisions [9]. To date, more than 250 definitions of health literacy exist in the literature, government reports, and organizational reports [10]. Broadly defined, it is an individual’s ability to access and understand essential health information and services and use them to make sound decisions for maintaining and promoting their health [11]. Given the understanding of the concept of health literacy, the ADA defines OHL as “the ability of an individual to obtain, process, and understand basic oral health information and services to make appropriate oral health strategies” [12].

 -      The objective of the manuscript does not correspond to what was presented throughout the text because no strategies or measures aimed at improving OHL were proposed. I strongly recommend the objective rewrite:

Therefore, the aim of this study was to extract the conceptual connotation of OHL based on a review of the concept in the relevant literature and provide a basis for proposing strategies and measures to improve OHL among Chinese residents.

-      It was unclear the weight coefficients calculation (Table 4).

-   Please, explain this affirmation below because it seems incorrect:

Good access to information promotes individual OHL improvement, which in turn lowers OHL levels”

-      Please review and align the manuscript with the Qualitative Research Checklist of Critical Appraisal Skills Programme (CASP) reporting guidelines.

 -      I suggest adding implications or suggestions arising from the results in the conclusion section.

-         Checked for duplicate references. Please review the references:

Macek MD, Haynes D, Wells W, Bauer-Leffler S, Cotten PA, Parker RM. Measuring conceptual health knowledge in the context of oral health literacy: preliminary results [J] . Journal of Public Health Dentistry, 2010, 70(3):197-204.

Reviewer 3 Report

The title should be shortened without repeating the same phrases twice in one sentence: "A conceptual model".." the conceptual composition method"

The abstract is very enigmatic and requires rewriting in a language understandable to the average recipient, for example, the authors write: "Personal factors include demographic characteristics and personal abilities and characteristics." what are the personal characteristics?

The last paragraph of the introduction is inconsistent with most of the introduction, in my opinion, it is confusing. Maybe the authors should adapt the title to the most well-written introduction, which could be "Chinese Population Oral Health Awareness - Effectiveness Models"

Section 2.1. Search strategies and screening criteria Remove duplicates of searching specific databases.

Section 2.2. Conceptual composition It should be shortened and simplified to "The conceptual diagram", explaining only what it consisted of without introducing unnecessary additional concepts, i.e.: Concept composition or concept map.

Subsections 2.2.1 to 2.2.4 should be shortened and placed as a caption under Figure No. 1.

Section 3.6 should be shortened and section 3.7 deleted.

Chapter 3.8 seems to be crucial for the subsections presented in the manuscript, the shortening or deletion of which should be considered. One summary table should be prepared for subsections: 3.8.1.-3.8.5.